# $\mathcal{P}$-BN: Towards Effective Batch Normalization in the Path Space

## Abstract

Neural networks with ReLU activation functions have demonstrated their success in many applications. Recently, researchers noticed a potential issue with the optimization of ReLU networks: the ReLU activation functions are positively scale-invariant (PSI), while the weights are not. This mismatch may lead to undesirable behaviors in the optimization process. Hence, some new algorithms that conduct optimizations directly in the *path space* (the path space is proven to be PSI) were developed, such as Stochastic Gradient Descent (SGD) in the path space, and it was shown that, SGD in the path space is superior to that in the weight space. However, it is still unknown that whether other deep learning techniques beyond SGD, such as batch normalization (BN), could also have their counterparts in the path space. In this paper, we conduct a formal study on the design of BN in the path space. According to our study, the key challenge is how to ensure the forward propagation in the path space, because BN is utilized during the forward process. To tackle such challenge, we propose a novel re-parameterization of ReLU networks, with which we replace each weight in the original neural network, with a new value calculated from one or several paths, while keeping the outputs of the network unchanged for any input. Then we show that BN in the path space, namely $\mathcal{P}$-BN, is just a slightly modified conventional BN on the re-parameterized ReLU networks. Finally, our experiments on two benchmark datasets, CIFAR and ImageNet, show that the proposed $\mathcal{P}$-BN can significantly outperform the conventional BN in the weight space.

## 1 Introduction

In recent years, neural networks with rectified linear unit (ReLU) activation functions (abbrev. ReLU networks) , have been successfully applied to many domains, such as image classification (He et al., 2016), game playing (Mnih et al., 2015), and text processing (Kim, 2014). Recently, it has been noticed that the feedforward neural networks with ReLU, namely ReLU networks, are positively scale-invariant (PSI) (Neyshabur et al., 2015a; 2016), which means when the incoming and outgoing weights of a hidden node, are respectively multiplied and divided by a positive constant, the outputs of ReLU networks will keep unchanged for any input. However, the vector space, composed of weights, in which the conventional optimization algorithms, e.g., Stochastic Gradient Descent (S-GD), optimize the neural networks, is not PSI, and such mismatch may lead to undesirable behaviors in the optimization process (Neyshabur et al., 2015a).

To tackle such challenge, some recent studies have shown that, one ReLU network can be optimized in a completely new PSI parameter space, i.e., the *path space*, instead of its original weight space (Meng et al., 2019). Detailedly, while regarding ReLU networks as directed acyclic graphs (DAGs), we can calculate the outputs of ReLU networks by using path-values, i.e., the multiplication of weights along each input-output path in the DAG. Here, path-values are invariant to the positive rescaling of weights, which exactly matches the PSI property of ReLU networks, and the optimization algorithms (e.g., SGD) in the path space (i.e., the vector space composed of path-values), are shown to be superior to those in the weight space (Neyshabur et al., 2015a; 2016; Meng et al., 2019).

There are already some optimization algorithms developed in the path space, however, to the best of our knowledge, it is still unknown whether other deep learning techniques beyond SGD, can also have their counterparts in the path space. For example, BN is one of the most widely used normal-

ization approaches (Ioffe & Szegedy, 2015; Santurkar et al., 2018), and is crucial for facilitating the training process of neural networks. With the help of BN, the training process of neural networks can be accelerated, by both decoupling the scale and direction of weight vectors (Arora et al., 2018) and smoothing the optimization landscape (Santurkar et al., 2018).

In this paper, we conduct a formal study on the design of BN in the path space. *First*, it is known that, BN is successfully utilized during the forward process in the weight space, but how to ensure the forward propagation still remains unclear in the path space, i.e., the way to calculate the outputs of hidden nodes via path-values layer by layer is unknown. For example, in Fig. 1, we show a network and its outputs calculated by both weights and path-values. Here, the outputs of this network can be simply calculated by using path-values, because a path starts from an input node and ends up with an output node, and the path-value is just the multiplication of weights along this path. Nevertheless, the outputs of hidden nodes cannot be calculated by path-values in this obvious way. Thus, to solve this problem, we propose a *re-parameterization* for the weights in ReLU networks, which can replace each weight in the

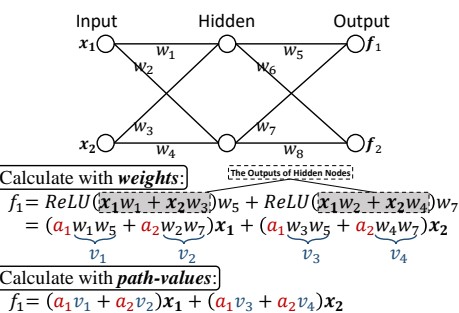

Figure 1: An example of an MLP network. $a_1$ and $a_2$ are 1 when corresponding ReLU is activated, otherwise they are 0.

original network, with a new value calculated by one or several path-values, and the outputs of the re-parameterized network will keep unchanged for any input. Therefore, the forward propagation in the path space can be easily conducted after re-parameterizing the network.

*Then*, we show that BN in the path space is just a slightly modified conventional BN on the re-parameterized network. Considering that BN in the weight space, can make the gradient propagation more effective, we study BN in the path space from the perspective of effective propagation. In details, we apply the conventional BN to the re-parameterized network, and find that the radients will explode during the backward propagation in the path space. Thus, we propose a novel BN in the path space, namely $\mathcal{P}$-*BN*, which slightly modifies the conventional BN, via excluding the term whose coefficient is fixed by re-parameterization from normalization. Particularly, we theoretically prove and experimentally show that compared with the conventional BN, $\mathcal{P}$-BN can propagate gradients in the path space more effectively.

*Finally*, we conduct extensive experiments on multiple datasets, including CIFAR, ImageNet, with multiple network structures, including ResNet and PlainNet. The results demonstrate that the $\mathcal{P}$-BN based optimization algorithms can significantly outperform the ones with the conventional BN.

## 2 BACKGROUND

### 2.1 RELATED WORK

In recent years, some researchers have conducted many theoretical studies on the *path* of ReLU networks. *On the one hand*, in terms of optimization, for example, Neyshabur et al. (2015a) proposed a new regularization term, i.e., path-norm, and the Path-SGD algorithm for minimizing the regularized loss. Meng et al. (2019) proposed $\mathcal{G}$-SGD algorithm to directly optimize the model in the path space, by utilizing the gradients w.r.t. path-values. *On the other hand*, in terms of generalization, for example, Neyshabur et al. (2015b) and Zheng et al. (2018) gave the relationship between generalization error and path-norm or basis path-norm, respectively. Besides, E et al. (2018) leveraged the path representation of ReLU networks, to analyze the generalization error of them.

However, the aforementioned studies have not involved normalization approaches (Ba et al., 2016; Wu & He, 2018), such as BN (Ioffe & Szegedy, 2015), which are proved to be much crucial for training neural networks. In particular, as for ReLU networks, BN is especially widely used in computer vision domain (He et al., 2016; Li et al., 2016; Huang et al., 2017). Furthermore, there are also many theoretical studies on BN (Santurkar et al., 2018; Arora et al., 2018; Kohler et al., 2019).

Accordingly, based on the studies on optimizing ReLU networks in the path space, and the crucial normalization technique, i.e., BN, in this paper, we will give a formal study on designing BN in the path space. We will next briefly introduce the optimization of ReLU networks without BN in the path space.

## 2.2 OPTIMIZING RELU NETWORKS IN THE PATH SPACE

In this subsection, we introduce the feedforward ReLU network and the optimization algorithm in the path space.

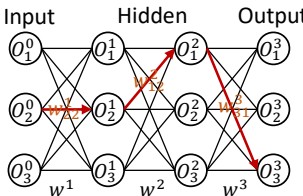

*Firstly*, as shown in Fig. 2, consider an $L$-layer feedforward ReLU network *without BN*[1] whose weight matrices are $\{w^l; l = 1, \cdots, L\}$, we use $O^l_j, l = 0, \cdots, L$ to denote the *j-th* node in the *l-th* layer. Based on these above, we further use $w^l_{jk}$ to represent the weight on the edge which connects $O^{l-1}_k$ and $O^l_j$. As for the outputs of hidden nodes, suppose that given a $d$-dimensional input $x = (x_1, \cdots, x_d)$, then the output of $O^l_j$ and the hidden vector in the *l-th* layer, can be denoted as $o^l_j(x)$ and $o^l(x)$, respectively. Thus, the outputs can be propagated as $o^l(x) = g(w^l o^{l-1}(x))$, where $g(\cdot) = max(\cdot, 0)$ is a ReLU activation function.

Figure 2: An example of a feedforward ReLU network.

*Secondly*, we will show the concept of a path in ReLU networks, and briefly introduce the optimization algorithm in the path space. In details, regarding the network structure as a DAG consisting of nodes and edges, a **path** can be defined as: a list of nodes or edges, starting from an input node, successively passing by several hidden nodes along the edges, and finally ending up with an output node. For example, in Fig. 2, path $p$ starts from $O^0_2$, passes $O^1_2$ and $O^2_1$, and ends up with $O^3_3$. And then, the **path-value** of path $p$ can be defined as a multiplication of the weights along $p$, i.e., $v_p = w^2_{22} w^2_{12} w^3_{31}$. Therefore, output $o^L_j(x)$ can be calculated by using path-values as $o^L_j(x) = \sum_{k=1}^d \sum_{p \in \mathbb{P}_{k,j}} v_p \cdot a_p \cdot x_k$. Here, $\mathbb{P}$ is a set which contains the paths staring from $O^0_k$ and ending up with $O^L_j$, and $a_p$, the ReLU activation status of path $p$, can be calculated as $a_p = \prod_{O^{l_p}_{j_p} \in p} \mathbb{I}(o^{l_p}_{j_p}(x) > 0)$, where $O^{l_p}_{j_p} \in p$ represents that $O^{l_p}_{j_p}$ is passed by $p$.

As for the optimization algorithm, we introduce two typical ones, i.e., Path-SGD (Neyshabur et al., 2015a) and $\mathcal{G}$-SGD (Meng et al., 2019). Path-SGD is to add a new regularizer, i.e., path-norm, into the loss function, and then heuristically utilizes a coordinate-wised solution for minimizing the path-norm regularized loss. It is shown to be superior to SGD for MLP, but with huge computational costs when applied to Convolutional Neural Networks (CNNs), due to the complicate correlations among path-values. Besides, Path-SGD only utilizes the PSI property to propose the regularizer, but cannot optimize the network in the path space. Comparatively, $\mathcal{G}$-SGD can solve the computation problem, by decoupling correlations between path-values. Besides, $\mathcal{G}$-SGD can optimize the network via updating path-values by using gradients w.r.t. them instead of weights. Thus, $\mathcal{G}$-SGD can serve as the SGD in the path space, and the detailed on $\mathcal{G}$-SGD are listed in Appendix C.

## 3 TO DESIGN BN IN THE PATH SPACE

In this section, we first introduce a re-parameterization process, to ensure the forward propagation in the path space, and calculate the outputs of hidden nodes by using path-values. Then, we will show that the conventional BN cannot effectively propagate the error signals in the path space, which calls us to design an effective BN in the path space.

## 3.1 RE-PARAMETERIZATION

In this subsection, we introduce a re-parameterization for the parameters of ReLU networks, which can replace each weight with a path-value or a ratio of path-values, and the outputs of the re-

---

[1]For ease of presentation, we only discuss the MLP with equal width, and in this paper, the analyses and algorithms can be applied to the MLP and CNN with unequal width (cf. Appendix D.1).

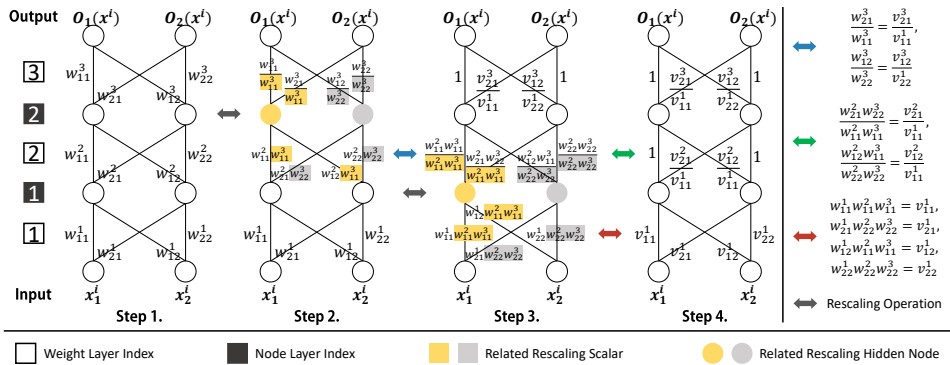

Figure 3: An example to demonstrate the re-parameterization process for an MLP.

parameterized network will keep unchanged for any input. First of all, we present the process of re-parameterization in Theorem 3.1.

**Theorem 3.1** *Consider an L-layer ReLU network with weight matrices $\{w^l; l = 1, \cdots, L\}$, if the diagonal elements in matrix $w^l, l > 2$ are positive, the hidden outputs of the ReLU network can be calculated by using path-values as follows:*

$$o_j^l(x) = g\left(\sum_{k=1}^d v_{jk}^l \cdot x_k\right) \quad for \; l = 1, \tag{1}$$

$$o_j^l(x) = g\left(o_j^{l-1}(x) + \sum_{k=1,k\neq j}^d \frac{v_{jk}^l}{v_{kk}^1} \cdot o_k^{l-1}(x)\right) \quad for \; l \geq 2. \tag{2}$$

*Here, $g(\cdot)$ denotes the ReLU activation function, and $v_{kk}^1$ is the path-value of the path which only contains the k-th diagonal elements in the weight matrices, i.e., $v_{kk}^1 = \prod_{l=1}^L w_{kk}^l$, and $v_{jk}^l, k \neq j$ is the path-value of the path which contains one non-diagonal weight, i.e., $v_{jk}^l = (\prod_{s<l} w_{kk}^s) \cdot w_{jk}^l \cdot (\prod_{s>l} w_{jj}^s)$.*

**Proof:** Based on the PSI property, we will prove Theorem 3.1 by designing a re-parameterization process which consists several weight rescaling steps. After these steps, the ReLU network can be equivalently re-parameterized, which means that the outputs will keep unchanged for any input after re-parameterization. Here, we will next demonstrate the re-parameterization process by showing that it can be conducted for a 3-layer MLP, as well as generalized to a multi-layer setting.

*As for a 3-layer MLP*, the process is shown in Fig. 3, and described in the following two steps: **(1)** The weights connected with nodes $O_j^2, j = 1, 2$ are rescaled, i.e., the incoming and outgoing weights of node $O_j^2$ are multiplied and divided by $w_{jj}^3$, respectively. To be specific, $w_{11}^3, w_{21}^3$ and $w_{11}^2, w_{12}^2$ are divided and multiplied by $w_{11}^3$, and similarly, $w_{22}^3, w_{12}^3$ and $w_{22}^2, w_{21}^2$ are also rescaled by using $w_{22}^3$. According to the definition of a path value, we have $\frac{w_{21}^3}{w_{11}^3} = \frac{w_{21}^3 w_{11}^2 w_{11}^1}{w_{11}^3 w_{11}^2 w_{11}^1} = \frac{v_{21}^3}{v_{11}^1}, \frac{w_{12}^3}{w_{22}^3} = \frac{w_{12}^3 w_{22}^2 w_{22}^1}{w_{22}^3 w_{22}^2 w_{22}^1} = \frac{v_{12}^3}{v_{22}^1}$, and hence, each weight in the *3rd* layer can be replaced by a constant or a ratio of path-values. **(2)** The weights connected with nodes $O_j^1, j = 1, 2$ are rescaled, i.e., the incoming and outgoing weights of node $O_j^1$ are multiplied and divided by the corresponding outgoing diagonal weight (i.e., $w_{jj}^2 w_{jj}^3$). Afterwards, each weight in the middle layer can be replaced by a constant or a ratio of path-values, and each weight in the *1st* layer can be replaced by a path-value. Therefore, each weight in this network are replaced by a path-value or a ratio of path-values.

*As for a multi-layer setting*, the re-parameterization process is similar to that of a 3-layer MLP. In particular, this process can be proceeded by rescaling weights orderly, from which is connected with the nodes in the last hidden node layer to the first hidden node layer, and the scalar of the rescaling operation at each node, is the corresponding outgoing diagonal weight. After such rescaling steps, the weights can be re-parameterized as follows: (1) $w_{jk}^1 \to v_{jk}^1$; (2) $w_{kk}^l \to 1$; (3) $w_{jk}^l \to \frac{v_{jk}^l}{v_{kk}^1}, l \neq 1, j \neq k$.

In summary, a network with weight at each edge (e.g., Step 1 in Fig. 3), can be equivalently re-parameterized into another network with a path-value or a ratio of path-values at each edge (e.g., Step 4 in Fig. 3), by rescaling its weights layer by layer. After the re-parameterization, the outputs of hidden nodes can be calculated by using path-values, in the same way as they are calculated by weights, and hence, Theorem 3.1 is established. ∎

**Remark**: The positive constraint is not essential for proving Theorem 3.1, because when the diagonal weights are negative, the re-parameterization can still be conducted correctly by using the absolute value of the corresponding weight as the rescaling scalar. Considering the robustness and computational efficiency, this constrain exists in the optimization algorithm (Meng et al., 2019), and hence, we follow it. Besides, this constrain will not bring much influence on the model expressiveness, because the number of the constrained weights is tiny, compared with the total weights. Moreover, we discuss some other choices on the re-parameterization and the influence in Appendix D.

## 3.2 ANALYZING THE BN IN THE PATH SPACE

With the re-parameterization in Section 3.1, the forward propagation can be ensured in the path space, and the outputs of hidden nodes can be calculated by using path-values. Hence, in this subsection, we will design BN in the path space, via applying and analyzing the conventional BN to the re-parameterized network.

*Firstly*, we briefly introduce the conventional BN. Given a mini-batch of inputs $\{x^i; i = 1, \cdots, m\}$, BN can normalize each output of the hidden node as $\mathcal{BN}(o_j^l(x^i)) = \gamma_j^l \cdot \frac{o_j^l(x^i) - \mu_j^l}{\sigma_j^l} + \beta_j^l$, where $\mu_j^l = \frac{1}{m}\sum_{i=1}^m o_j^l(x^i)$ and $\sigma_j^l = \sqrt{\frac{1}{m}\sum_{i=1}^m (o_j^l(x^i) - \mu_j^l)^2 + \epsilon}$ represent the mean and standard deviation of the minibatch outputs, and $\gamma_j^l, \beta_j^l$ are the scale and shift term [2] which are independent with $w$, respectively. Then, we can denote the output after BN and ReLU as $z_j^l(x^i) = g\left(\mathcal{BN}(o_j^l(x^i))\right)$.

*Secondly*, we analyze the gradient propagation in the re-parameterized network with the the conventional BN. Here, we use $\mathcal{L}$ and $\nabla_{z^l(x^i)}\mathcal{L} = \left(\frac{\partial \mathcal{L}}{\partial z_1^l(x^i)}, \cdots, \frac{\partial \mathcal{L}}{\partial z_d^l(x^i)}\right)$, to denote the cross-entropy loss and the gradient vector w.r.t hidden output in layer $l$, respectively. Then, we can give an estimation of the norm of $\nabla_{z^l(x^i)}\mathcal{L}$ in Theorem 3.2.

**Theorem 3.2** *We use $W_l'$ to denote the parameter matrix in layer $l$ after the re-parameterization. Suppose $z_j^l(x^i) < \mathcal{O}(1), \forall i, j, l$, and $m > 64$. The norm of the gradient w.r.t $z^l(x^i)$ can be upper bounded as*

$$\|\nabla_{z^l(x^i)}\mathcal{L}\| \leq \mathcal{O}\left(\prod_{s=l+1}^L \|D^s(x^i) \cdot \gamma^s \cdot (\sigma^s)^{-1} \cdot W_s'\|\right),$$

*where $\sigma^l = diag(\sigma_1^l, \cdots, \sigma_d^l)$, $\gamma^l = diag(\gamma_1^l, \cdots, \gamma_d^l)$, and $D^l(x^i) = diag(D_1^l(x^i), \cdots, D_d^l(x^i))$. Here, $diag(a_1, \cdots, a_d)$ represents a diagonal matrix whose diagonal elements are $a_1, \cdots, a_d$, and $D_j^l(x^i) = 1$ if $z_j^l(x^i) > 0$, otherwise $D_j^l(x^i) = 0$.*

The proof of Theorem 3.2 can be found in Appendix B.

**Discussion:** According to Theorem 3.2, the gradient norm exponentially depends on depth $L$. If the spectral norm of matrix $D^s(x^i) \cdot \gamma^s \cdot (\sigma_s)^{-1} \cdot W_s'$ roughly equals to 1, the error signals can be effectively propagated in the backward process (i.e., not vanished or exploded). On the one hand, if there is non-zero elements in $D^s(x^i)$, its spectral norm will equal to 1. On the other hand, in practice, $\gamma_j^s$ is initialized to be 1, and will keep its value around 1 during the training process, so the spectral norm of $\gamma^s$ will be around 1. Hence, the magnitudes of $W_s'$ and $\sigma_s^{-1}$ is crucial for the gradient norm. For matrix $W_s'$, according to Theorem 3.1, the diagonal elements of $W_s'(s \geq 2)$ are degenerated to be constant 1, and these constant terms will not be trained. Besides, the trained parameters are initialized to be much smaller than 1 (cf. Appendix D.3), so matrix $W_s'$ will approach to an identity matrix. For matrix $\sigma_s^{-1}$, if each $\sigma_j^s$ is smaller than 1, it can cause gradient exploding during the back propagation process. We will conduct observational experiments in Section 5.2, and it shows that $\|\sigma^s\|$ is less than 1 and gradients becomes larger during the back propagation process.

---

[2]In the following analysis, referring to Santurkar et al. (2018), we assume that both of the scale and shift term are constant.

Therefore, such discussion result motivates us to design BN in the path space by modifying the conventional BN, in order to propagate the error signals effectively. We will next detailedly introduce the proposed BN in the path space, i.e., $\mathcal{P}$-BN.

## 4 $\mathcal{P}$-BN: EFFECTIVE BN IN THE PATH SPACE

In this section, we design the BN in the path space, namely $\mathcal{P}$-BN, via slightly modifying the forward process of the conventional BN. Motivated by the calculation for the outputs of hidden node in Theorem 3.1, as well as the discussion of Theorem 3.2, we propose to normalize the terms related to the trained coefficients (i.e., Eq. 1 and the *2nd* term in Eq. 2), and exclude the term related to the constant coefficient (i.e., the *1st* term in Eq. 2). Specifically, we use $\bar{\bar{z}}_j^l(x^i)$ to denote the output after $\mathcal{P}$-BN and ReLU, and detailedly describe the forward process of $\mathcal{P}$-BN in the following two steps.

**(1)** For the *1st* layer, the operation of $\mathcal{P}$-BN is the same as the conventional BN, i.e., $\bar{\bar{z}}_j^l(x^i) = z_j^l(x^i)$.

**(2)** For the *l-th* layer ($l \geq 2$), as shown in Fig. 4, there are three sub-steps for $\mathcal{P}$-BN as follows.

*First*, calculate the partial weighted summation. Here, the input related to the diagonal constant element in the parameter matrix is excluded, i.e.,

$$o_j^{l,/}(x^i) = \sum_{k=1, k\neq j}^{d} \frac{v_{jk}^l}{v_{kk}^1} \cdot \bar{\bar{z}}_k^{l-1}(x) \tag{3}$$

*Second*, normalize the partial weighted summation above, i.e.,

$$\mathcal{BN}(o_j^{l,/}(x^i)) = \gamma_j^{l,/} \cdot \frac{o_j^{l,/}(x^i) - \mu_j^{l,/}}{\sigma_j^{l,/}} + \beta_j^{l,/} \tag{4}$$

*Third*, add the excluded term $\bar{\bar{z}}_j^{l-1}(x^i)$ into the normalized partial weighted summation in Eq. 4, i.e.,

$$\bar{\bar{z}}_j^l(x^i) = g\left(\mathcal{BN}(o_j^{l,/}(x^i)) + \bar{\bar{z}}_j^{l-1}(x^i)\right) \tag{5}$$

Figure 4: An example of the *2nd* step in $\mathcal{P}$-BN.

Consequently, $\mathcal{P}$-BN can be easily combined with the optimization algorithm in the path space for ReLU networks, and we will show it in Algorithm 1 in Appendix E.

Then, we will show that the gradients can be effectively propagated by using $\mathcal{P}$-BN in Theorem 4.1.

**Theorem 4.1** *Suppose* $\bar{\bar{z}}_j^l(x^i) < \mathcal{O}(1), \forall i, j, l$, *and* $m > 64$, *the norm of gradient w.r.t* $\bar{\bar{z}}^l(x^i)$ *can be upper bounded as*

$$\|\nabla_{\bar{\bar{z}}^l(x^i)} \mathcal{L}\| \leq \mathcal{O}\left(\prod_{s=l+1}^{L} \|\bar{\bar{D}}^s(x^i) \cdot (I + \gamma^{s,/} \cdot (\sigma^{s,/})^{-1} \cdot \hat{W}_s')\|\right),$$

*where* $\hat{W}_s' = W_s' - I$, $\sigma^{l,/} = diag(\sigma_1^{l,/}, \cdots, \sigma_d^{l,/})$, $\gamma^{l,/} = diag(\gamma_1^{l,/}, \cdots, \gamma_d^{l,/})$ *and* $\bar{\bar{D}}^l(x^i) = diag(\bar{\bar{D}}_1^l(x^i), \cdots, \bar{\bar{D}}_d^l(x^i))$ *where* $\bar{\bar{D}}_j^l(x^i) = 1$ *if* $\bar{\bar{z}}_j^l(x^i) > 0$, *otherwise* $\bar{\bar{D}}_j^l(x^i) = 0$.

The proof of Theorem 4.1 can be found in Appendix B. Then, we give Corollary 4.2, via summarizing from Theorem 3.2 and Theorem 4.1.

**Corollary 4.2** *Using the notations in Theorem 3.2 and Theorem 4.1 and initializing* $\gamma^l$ *and* $\gamma^{l,/}$ *to be identity matrix, we have the following upper bound for gradient norm at initialization point*

$$\|\nabla_{z^l(x^i)} \mathcal{L}\| \leq \mathcal{O}\left(\prod_{s=l+1}^{L} \|D^s(x^i) \cdot (\sigma^s)^{-1} \cdot (I + \hat{W}_s')\|\right),$$

$$\|\nabla_{\bar{\bar{z}}^l(x^i)} \mathcal{L}\| \leq \mathcal{O}\left(\prod_{s=l+1}^{L} \|\bar{\bar{D}}^s(x^i) \cdot \left(I + (\sigma^{s,/})^{-1} \cdot \hat{W}_s'\right)\|\right).$$

| Dataset | Method | PlainNet | | ResNet | | |
|---------|--------|----------|--|--------|--|--|
| | | 18 | 34 | 18 | 34 | 50 |
| CIFAR-10 | SGD+BN | 6.93% (±0.12) | 7.76% (±0.22) | 6.76% (±0.10) | 6.40% (±0.09) | 6.55% (±0.24) |
| | $\mathcal{G}$-SGD+BN | 6.66% (±0.12) | 6.74% (±0.13) | 6.84% (±0.30) | 6.54% (±0.06) | 6.62% (±0.19) |
| | $\mathcal{G}$-SGD+BN(wnorm) | 6.26% (±0.17) | 6.67% (±0.26) | 6.31% (±0.14) | 6.33% (±0.15) | 6.31% (±0.14) |
| | $\mathcal{G}$-SGD+$\mathcal{P}$-BN$^{ours}$ | **5.99% (±0.13)** | **6.04% (±0.16)** | **6.04% (±0.14)** | **5.66% (±0.10)** | **5.99% (±0.12)** |
| CIFAR-100 | SGD+BN | 28.10% (±0.19) | 33.37% (±0.41) | 26.97% (±0.12) | 26.42% (±0.23) | 25.62% (±0.18) |
| | $\mathcal{G}$-SGD+BN | 27.08% (±0.35) | 28.19% (±0.35) | 27.13% (±0.39) | 26.61% (±0.10) | 25.99% (±0.40) |
| | $\mathcal{G}$-SGD+BN(wnorm) | 26.76% (±0.05) | 27.61% (±0.24) | 26.60% (±0.17) | 26.72% (±0.27) | 25.65% (±0.22) |
| | $\mathcal{G}$-SGD+$\mathcal{P}$-BN$^{ours}$ | **26.70% (±0.10)** | **27.04% (±0.12)** | **26.39% (±0.10)** | **26.24% (±0.29)** | **25.29% (±0.19)** |

Table 1: Test error rate on CIFAR-10 and CIFAR-100.

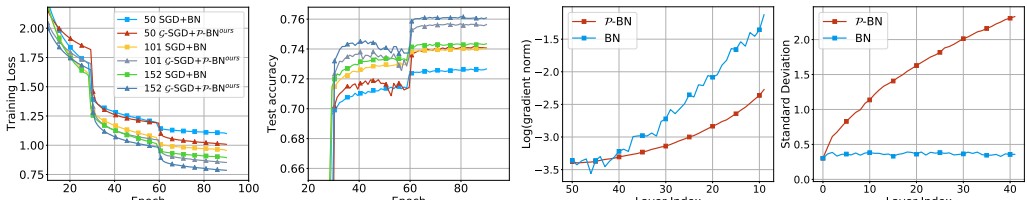

Figure 5: Curves for training ImageNet.    Figure 6: Experimental observations.

*Suppose $\lambda = \|\hat{W}'_s\|$. A sufficient condition for that $\|\nabla_{z^l(x^i)}\mathcal{L}\|$ and $\|\nabla_{\overline{z}^l(x^i)}\mathcal{L}\|$ can be upper bounded by constant (i.e., will not diverge as $L \to \infty$) is $\sigma_j^s \in [\frac{L(1+\lambda)}{L+1}, \frac{L(1+\lambda)}{L-1}]$ and $\sigma_j^{s,/} \in [\lambda L, \infty)$ $\forall j$, respectively.* [3]

**Discussion:** We can conclude from Corollary 4.2 that the range for $\sigma_j^{s,/}$ is much larger than that for $\sigma_j^s$. Furthermore, if $\lambda \leq \frac{1}{L}$, we have $[\frac{L(1+\lambda)}{L+1}, \frac{L(1+\lambda)}{L-1}] \subset [\lambda L, \infty)$ (i.e., the range for $\sigma_j^s$ is fully contained in the range $\sigma_j^{s,/}$), which indicates the backward stability of $\mathcal{P}$-BN. We will observe the variance in each layer in Section 5.2, and the results demonstrate that the variance becomes larger with $\mathcal{P}$-BN. Thus, to some extent, the gradient propagated more stable for NN with $\mathcal{P}$-BN.

## 5 EXPERIMENTS

In this section, we conduct experiments by first comparing the performance of $\mathcal{P}$-BN with the baselines on various datasets with some network structures, and then showing some experimental observations to support theoretical analyses.

### 5.1 PERFORMANCE EXPERIMENTS

In this subsection, we evaluate the performance of P-BN on training deep neural networks by conducting experiments on three datasets, CIFAR-10, CIFAR-100, and ImageNet. We will next introduce the network structures and compared methods first.

#### 5.1.1 NETWORK STRUCTURES AND COMPARED METHODS

In this subsection, we describe the network structures and compared methods. *First*, as for the network structures, we apply $\mathcal{P}$-BN to train ResNet and PlainNet (He et al., 2016). *Second*, we show the setting details of four compared methods as follows.

*(1)* SGD+BN: We use SGD to train the network with the conventional BN.
*(2)* $\mathcal{G}$-SGD+BN: We use $\mathcal{G}$-SGD to train the network with the conventional BN.
*(3)* $\mathcal{G}$-SGD+BN(wnorm): We use a method which was intuitively proposed to handle the conventional BN without any theoretical analysis (Meng et al., 2019). Here, the gradient of the path-value,

---

[3]Please note that the conditions are tight in the sense that if we change $L$ in the ranges to be $L^{1-\epsilon}$, the upper bound of both the gradient norm for BN and $\mathcal{P}$-BN will diverge.

is normalized by the L2-norm of the weights pointing to the same hidden unit.

*(4) $\mathcal{G}$-SGD+$\mathcal{P}$-BN:* Our proposed $\mathcal{P}$-BN targets on BN in the path space, and the network with $\mathcal{P}$-BN should be optimized by the algorithms in the path space. Thus, we use $\mathcal{G}$-SGD to train the network with $\mathcal{P}$-BN.

### 5.1.2 CIFAR

We conduct experiments on CIFAR-10 dataset and CIFAR-100 dataset (Krizhevsky et al., 2009). Here, we train ResNet of 18, 34, and 50 layers, and PlainNet of 18 and 34 layers, respectively. As for the hyper-parameters of methods mentioned in their corresponding papers, i.e., SGD+BN (He et al., 2016) and $\mathcal{G}$-SGD+BN(wnorm) (Meng et al., 2019), we use the same settings as their original ones. Besides, we tune the hyper-parameters for other methods, i.e., $\mathcal{G}$-SGD+BN, and the proposed $\mathcal{G}$-SGD+$\mathcal{P}$-BN. More details about the experimental setting can refer to Appendix F.1. In Table 1, we

| Metric | ResNet | Method | |
|---|---|---|---|
| | | SGD+BN | $\mathcal{G}$-SGD+$\mathcal{P}$-BN |
| Top 1 Test Error Rate | 50 | 27.34% ($\pm$0.10) | **25.92%** ($\pm$**0.18**) |
| | 101 | 25.94% ($\pm$0.09) | **24.36%** ($\pm$**0.10**) |
| | 152 | 25.65% ($\pm$0.15) | **23.91%** ($\pm$**0.23**) |
| Top 5 Test Error Rate | 50 | 9.10% ($\pm$0.06) | **8.26%** ($\pm$**0.16**) |
| | 101 | 8.33% ($\pm$0.07) | **7.43%** ($\pm$**0.05**) |
| | 152 | 8.31% ($\pm$0.09) | **7.25%** ($\pm$**0.09**) |

Table 2: Test error rate on ImageNet.

show the performance results on the test error rate, and in Fig. 8, we plot the training curve and test accuracy of 50-layer ResNet. Such results demonstrate that: *(1)* $\mathcal{G}$-SGD+$\mathcal{P}$-BN outperforms others on all tested datasets and network structures, which shows the superiority of $\mathcal{P}$-BN; *(2)* Combining the path space and the conventional BN directly hurts performance, which empirically motivates us to propose $\mathcal{P}$-BN; *(3)* $\mathcal{G}$-SGD+$\mathcal{P}$-BN outperforms $\mathcal{G}$-SGD+BN(wnorm), which shows the benefit and significance of our proposed formal analyses.

### 5.1.3 IMAGENET

We conduct experiments on ImageNet dataset (Russakovsky et al., 2015). Here, we train ResNet of 50, 101 and 152 layers, to demonstrate that the proposed $\mathcal{P}$-BN can help to train very deep networks on a large dataset, and we compare the performance results between SGD+BN and $\mathcal{G}$-SGD+$\mathcal{P}$-BN. The method for tuning the hyper-parameters is similar to that in Section 5.1.2, and more details can refer to Appendix F.1. In Table 2, we list top-1 and top-5 test error rates, and in Fig. 5, plot the training curve and test accuracy during training ResNet of 50, 101, and 152 layers. These results well demonstrate the superiority of $\mathcal{G}$-SGD+$\mathcal{P}$-BN, and show that for the large dataset and very deep networks, $\mathcal{P}$-BN can still outperform the conventional BN.

### 5.2 OBSERVATIONS ON STANDARD DEVIATIONS AND GRADIENTS

After analyzing the advantage of $\mathcal{P}$-BN over the conventional BN theoretically in Section 3.2 and 4, we now provide some experimental observations to support our analyses. Here, we will show the magnitude of standard deviations in BN, and the gradients of ReLU networks with BN and $\mathcal{P}$-BN.

In this experiments, we use a stacked CNN of 50 layers with 128 channels in each layer, and use CIFAR-10 as training data, with the batch size of 128. Then, we add the conventional BN or $\mathcal{P}$-BN at the end of each hidden layer in the stacked CNN, which is denoted as *BN* and *$\mathcal{P}$-BN*, respectively. For the model with the conventional BN, it is randomly initialized. For the model with $\mathcal{P}$-BN, the diagonal elements in parameter matrices (expect for the *1st* layer) are initialized to 1 (cf. Section 3.1), and other elements are initialized with small random values (cf. Appendix D.3). Because the theoretical analyses are not related to the training process, we will just observe some quantities at initialization as follows.

*First*, we show some observations on standard deviation $\sigma_j^l$ in Theorem 3.2. Here, we log $\sigma_j^l$ for each hidden node during the forward process of the *1st* mini-batch, and then calculate the average standard deviation among the nodes in each layer. In Fig. 6, we plot the curves regarding standard deviations, and we can observe that: *(1)* For the conventional BN, the standard deviations are smaller than 1, which matches the claim in Theorem 3.2, and is one of the reasons of the gradient exploding. *(2)* For $\mathcal{P}$-BN, the standard deviation grows when the layer index becomes larger, and most of them are larger than 1. Thus, in Theorem 4.1, $\|\sigma_s^{-1} \cdot W_s'\|$ are smaller than $\|I\|$, and $\|\nabla_{z^l}\mathcal{L}\|$ can be upper bounded, which is one of the reasons for that the gradient exploding can be weaken by $\mathcal{P}$-BN.

*Second*, we show some observations on gradients, to investigate that whether the gradient exploding can be weaken by $\mathcal{P}$-BN in practice. Here, we log the L2-norm of gradients of loss w.r.t. the outputs in each layer, and plot the log of the gradient norm in Fig. 6. We can observe that, the gradients indeed explode in the ReLU network with both the conventional BN and $\mathcal{P}$-BN, and the exploding speed is much slower with $\mathcal{P}$-BN, which demonstrates that $\mathcal{P}$-BN can help to weaken the gradient exploding.

## 6 CONCLUSION

In this paper, we conduct a formal study on the design of BN in the path space. First, considering that BN is adopted during the forward process, we ensure the forward propagation in the path space, via proposing a re-parameterization for weights in ReLU networks. Here, the re-parameterization can equivalently replace each weight in the original network with a new value calculated by one or several path-values, and hence, the outputs of hidden nodes can be calculated by using path-values. Then, we design BN in the path space, by applying and analyzing the conventional BN to the re-parameterized network. Here, we analyze the gradient propagation after BN is applied, and then notice that the gradient will explode in such networks. Thus, we propose a novel BN in the path space, i.e., $\mathcal{P}$-BN, which excludes the term whose coefficient is fixed by re-parameterization from normalization. Finally, we conduct experiments to verify the proposed method, and results on both CIFAR and ImageNet dataset, demonstrate that $\mathcal{P}$-BN can enhance the performance significantly.

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

# A  NOTATIONS

| Notation | Object |
|---|---|
| $x$ | $d$-dimensional input, i.e., $x = (x_1, \cdots, x_d)$ |
| $m$ | Minibatch size |
| $w^l$ | Weight matrix in the $l$-th layer |
| $o_j^l(x)$ | Output of the $j$-th hidden node in the $l$-th layer |
| $o^l(x)$ | Hidden vector in the $l$-h layer |
| $w_{jk}^l$ | Weight on the edge which connects the $k$-th node in the $(l-1)$-th layer and the $j$-th node in the $l$-th layer |
| $v_{kk}^1$ | Path-value of the path which only contains the $k$-th diagonal elements in weight matrices, i.e., $v_{kk}^1 = \prod_{l=1}^{L} w_{kk}^l$ |
| $v_{jk}^l, k \neq j$ | Path-value of the path which contains one non-diagonal weight, i.e., $v_{jk}^l = (\prod_{s<l} w_{kk}^s) \cdot w_{jk}^l \cdot (\prod_{s>l} w_{jj}^l)$ |
| $z_j^l(x)$ | Output after BN and ReLU of the $j$-th hidden node in the $l$-th layer |
| $\overline{\overline{z}}_j^l(x)$ | Output after $\mathcal{P}$-BN and ReLU of the $j$-th hidden node in the $l$-th layer |
| $W_l'$ | Parameter matrix in layer $l$ after the re-parameterization. |
| $\mu_j^l$ | Mean of the minibatch outputs of the $j$-th hidden node in the $l$-th layer |
| $\sigma_j^l$ | Standard deviation of the minibatch outputs of the $j$-th hidden node in the $l$-th layer |

Table 3: Main notations in this paper.

# B  SUPPLEMENTARY PROOFS

**Theorem B.1** *We use $W_l'$ to denote the parameter matrix in layer $l$ after the re-parameterization. Suppose $z_j^l(x^i) < \mathcal{O}(1), \forall i, j, l$, and the batch size $m > 64$. The norm of gradient w.r.t $z^l(x^i)$ can be upper bounded as*

$$\|\nabla_{z^l(x^i)} \mathcal{L}\| \leq \mathcal{O}\left( \prod_{s=l+1}^{L} \|D^s(x^i) \cdot \gamma^s \cdot (\sigma^s)^{-1} \cdot W_s'\| \right),$$

*where $\sigma^l = diag(\sigma_1^l, \cdots, \sigma_d^l)$, $\gamma^l = diag(\gamma_1^l, \cdots, \gamma_d^l)$ and $D^l(x^i) = diag(D_1^l(x^i), \cdots, D_d^l(x^i))$. Here, $diag(a_1, \cdots, a_d)$ represents a diagonal matrix whose diagonal elements are $a_1, \cdots, a_d$, and $D_j^l(x^i) = 1$ if $z_j^l(x^i) > 0$, otherwise $D_j^l(x^i) = 0$.*

**Proof:** We concat the gradient vector of different instances in a minibatch into one $(d \times m)$-dimensional vector as

$$\nabla_{z^l} \mathcal{L} = (\nabla_{z^l(x^1)} \mathcal{L}, \cdots \nabla_{z^l(x^m)} \mathcal{L}), \tag{6}$$

and hence, according to chain rule, we have

$$\nabla_{z^l} \mathcal{L} = \nabla_{z^L} \mathcal{L} \cdot \frac{\partial z^L}{\partial o^L} \cdot \frac{\partial o^L}{\partial z^{L-1}} \cdots \frac{\partial o^{l+1}}{\partial z^l}. \tag{7}$$

Here, on the one hand, as for matrix $\frac{\partial z^s}{\partial o^s}$, it donates

$$\frac{\partial z^s}{\partial o^s} = \begin{bmatrix} \frac{\partial z_1^s(x^1)}{\partial o_1^s(x^1)} & \frac{\partial z_1^s(x^1)}{\partial o_2^s(x^1)} & \cdots & \frac{\partial z_1^s(x^1)}{\partial o_d^s(x^m)} \\ \frac{\partial z_2^s(x^1)}{\partial o_1^s(x^1)} & \frac{\partial z_2^s(x^1)}{\partial o_2^s(x^1)} & \cdots & \frac{\partial z_2^s(x^1)}{\partial o_d^s(x^m)} \\ & \cdots & \cdots & \\ \frac{\partial z_d^s(x^m)}{\partial o_1^s(x^1)} & \frac{\partial z_d^s(x^m)}{\partial o_2^s(x^1)} & \cdots & \frac{\partial z_d^s(x^m)}{\partial o_d^s(x^m)} \end{bmatrix}. \tag{8}$$

So we have

$$\frac{\partial z_j^s(x^i)}{\partial o_j^s(x^i)} = D_j^s(x^i) \cdot \frac{\gamma_j^s}{\sigma_j^s}(1 - \frac{1}{m} - \frac{1}{m} \cdot (z_j^s(x^i))^2), \quad \text{if } j_1 = j_2 = j \text{ and } i_1 = i_2 = i; \quad (9)$$

$$\frac{\partial z_j^s(x^{i_1})}{\partial o_j^s(x^{i_2})} = D_j^s(x^{i_1}) \cdot \frac{\gamma_j^s}{\sigma_j^s}(-\frac{1}{m} - \frac{1}{m} \cdot z_j^s(x^{i_1}) \cdot z_j^s(x^{i_2})), \quad \text{if } j_1 = j_2 = j \text{ and } i_1 \neq i_2; \quad (10)$$

$$\frac{\partial z_{j_1}^s(x^{i_1})}{\partial o_{j_2}^s(x^{i_2})} = 0, \quad \text{if } j_1 \neq j_2. \quad (11)$$

Obviously, most of the elements in the matrix equals to zero. Under the assumption that $z_j^s(x^i) < \mathcal{O}(1)$ for all $i, j, l$ and $m > 64$, the above matrix approaches to the diagonal matrix $D^s \cdot \gamma^s \cdot (\sigma^s)^{-1}$.

On the other hand, as for matrix $\frac{\partial o^s}{\partial z^{s-1}}$, it donates

$$\frac{\partial o^s}{\partial z^{s-1}} = \begin{bmatrix} W_s' & 0 & \cdots & 0 \\ 0 & W_s' & \cdots & 0 \\ & & \cdots & \\ 0 & 0 & \cdots & W_s' \end{bmatrix} \quad (12)$$

Thus, using the fact that $\|AB\| \leq \|A\|\|B\|$, we can get the result in the theorem. ∎

**Theorem B.2** *Suppose $\overline{\overline{z}}_j^l(x^i) < \mathcal{O}(1), \forall i, j, l,$ and $m > 64$. The norm of gradient w.r.t $\overline{\overline{z}}^l(x^i)$ can be upper bounded as*

$$\|\nabla_{\overline{\overline{z}}^l(x^i)}\mathcal{L}\| \leq \mathcal{O}\left(\prod_{s=l+1}^{L} \|\overline{\overline{D}}^s(x^i) \cdot (I + \gamma^{s,/} \cdot (\sigma^{s,/})^{-1} \cdot \hat{W}_s')\|\right),$$

*where $\hat{W}_s' = W_s' - I$, $\sigma^{l,/} = diag(\sigma_1^{l,/}, \cdots, \sigma_d^{l,/})$, $\gamma^{l,/} = diag(\gamma_1^{l,/}, \cdots, \gamma_d^{l,/})$ and $\overline{\overline{D}}^l(x^i) = diag(\overline{\overline{D}}_1^l(x^i), \cdots, \overline{\overline{D}}_d^l(x^i))$ where $\overline{\overline{D}}_j^l(x^i) = 1$ if $\overline{\overline{z}}_j^l(x^i) > 0$, otherwise $\overline{\overline{D}}_j^l(x^i) = 0$.*

**Proof:** According to the chain rule, we have

$$\nabla_{\overline{\overline{z}}^l}\mathcal{L} = \nabla_{\overline{\overline{z}}^L}\mathcal{L} \cdot \left(\frac{\partial \overline{\overline{z}}^L}{\partial o^{L,/}} \cdot \frac{\partial o^{L,/}}{\partial \overline{\overline{z}}^{L-1}} + \frac{\partial \overline{\overline{z}}^L}{\partial \overline{\overline{z}}^{L-1}}\right) \cdots \left(\frac{\partial \overline{\overline{z}}^{l+1}}{\partial o^{l+1,/}} \cdot \frac{\partial o^{l+1,/}}{\partial \overline{\overline{z}}^l} + \frac{\partial \overline{\overline{z}}^{l+1}}{\partial \overline{\overline{z}}^l}\right), \quad (13)$$

where $\frac{\partial \overline{\overline{z}}^s}{\partial o^{s,/}}$ denotes

$$\frac{\partial \overline{\overline{z}}^s}{\partial o^{s,/}} = \begin{bmatrix} \frac{\partial \overline{\overline{z}}_1^s(x^1)}{\partial o_1^{s,/}(x^1)} & \frac{\partial \overline{\overline{z}}_1^s(x^1)}{\partial o_2^{s,/}(x^1)} & \cdots & \frac{\partial \overline{\overline{z}}_1^s(x^1)}{\partial o_d^{s,/}(x^m)} \\ \frac{\partial \overline{\overline{z}}_2^s(x^1)}{\partial o_1^{s,/}(x^1)} & \frac{\partial \overline{\overline{z}}_2^s(x^1)}{\partial o_2^{s,/}(x^1)} & \cdots & \frac{\partial \overline{\overline{z}}_2^s(x^1)}{\partial o_d^{s,/}(x^m)} \\ & \cdots & \cdots & \\ \frac{\partial \overline{\overline{z}}_d^s(x^m)}{\partial o_1^{s,/}(x^1)} & \frac{\partial \overline{\overline{z}}_d^s(x^m)}{\partial o_2^{s,/}(x^1)} & \cdots & \frac{\partial \overline{\overline{z}}_d^s(x^m)}{\partial o_d^{s,/}(x^m)} \end{bmatrix}. \quad (14)$$

So we have

$$\frac{\partial \overline{\overline{z}}_j^s(x^i)}{\partial o_j^{s,/}(x^i)} = \overline{\overline{D}}_j^s(x^i) \cdot \frac{\gamma_j^{s,/}}{\sigma_j^{s,/}}(1 - \frac{1}{m} - \frac{1}{m} \cdot (\mathcal{BN}(o_j^{s,/}(x^i))^2), \quad \text{if } j_1 = j_2 = j \text{ and } i_1 = i_2 = i;$$
$$(15)$$

$$\frac{\partial \overline{\overline{z}}_j^s(x^{i_1})}{\partial o_j^{s,/}(x^{i_2})} = \overline{\overline{D}}_j^s(x^{i_1}) \cdot \frac{\gamma_j^{s,/}}{\sigma_j^{s,/}}(-\frac{1}{m} - \frac{1}{m} \cdot \mathcal{BN}(o_j^{s,/}(x^{i_1})) \cdot \mathcal{BN}(o_j^{s,/}(x^{i_2}))),$$

$$\text{if } j_1 = j_2 = j \text{ and } i_1 \neq i_2; \quad (16)$$

$$\frac{\partial \overline{\overline{z}}_{j_1}^s(x^{i_1})}{\partial o_{j_2}^{s,/}(x^{i_2})} = 0, \quad \text{if } j_1 \neq j_2. \quad (17)$$

Obviously, most of the elements in the matrix equals to zero. Under the assumption that $\overline{\overline{z}}_j^l(x^i)$ is bounded for all $i, j, l$, the above matrix tend to be the diagonal matrix $D^l \cdot \gamma^l \cdot (\sigma^l)^{-1}$ if $m$ approaches $\infty$.

On the one hand, as for matrix $\frac{\partial o^{s,l}}{\partial \overline{\overline{z}}^{s-1}}$, we have

$$\frac{\partial o^{s,l}}{\partial \overline{\overline{z}}^{s-1}} = I. \tag{18}$$

On the other hand, as for matrix $\frac{\partial o^{s,l}}{\partial \overline{\overline{z}}^{s-1}}$, we have

$$\frac{\partial o^{s,l}}{\partial \overline{\overline{z}}^{s-1}} = \begin{bmatrix} \hat{W}'_s & 0 & \cdots & 0 \\ 0 & \hat{W}'_s & \cdots & 0 \\ & & \cdots & \\ 0 & 0 & \cdots & \hat{W}'_s \end{bmatrix} \tag{19}$$

Thus, using the fact that $\|AB\| \le \|A\|\|B\|$, we can get the result in the theorem. ∎

## C    DETAILS ON OPTIMIZATION ALGORITHMS IN THE PATH SPACE

We provide the detailed update rules on $\mathcal{G}$-SGD (Meng et al., 2019) in this section. Here, we use $w_j$ and $w_i$ to denote the diagonal weight and non-diagonal weight at layer 1, respectively, and use $w_{j'}$ and $w_{i'}$ to denote the diagonal weight and non-diagonal weight at layer $l$, $l \ge 2$, respectively. The gradient of $w$ w.r.t loss $l$, can be represented by $\delta_w$. Then we have

$$w_j^{t+1} = w_j^t - \eta_t \cdot \frac{\delta_{w_j}^t \cdot w_j^t - \cdot \sum_{w_j \to w_{i'}} \delta_{w_{i'}}^t \cdot w_{i'}^t}{w_j^t}, \quad w_{j'}^{t+1} = w_{j'}^t; \tag{20}$$

$$w_i^{t+1} = w_i^t - \eta_t \cdot \delta_{w_i}^t, \quad w_{i'}^{t+1} = \frac{w_{i'}^t - \eta_t \cdot \delta_{w_{i'}}^t / (w_j^t)^2}{\mathbb{I}(w_j \to w_{i'}) \cdot w_j^{t+1} / w_j^t}, \tag{21}$$

where event $w_j \to w_{i'}$, means $w_j$ and $w_{i'}$ are contained in one of the paths, containing one non-diagonal weights at most.

## D    DISCUSSION ABOUT RE-PARAMETERIZATION

### D.1    UNEQUAL WIDTH MLP AND CNN

In this section, we first show the method for finding weights used for rescaling in Theorem 3.1, which is denoted as "red weights" here, for the network with unequal width of the input and output layers.

As shown in Fig. 7, for weight matrix $\{w^l; l = 2, \cdots, L-1\}$, the diagonal elements are selected as red weights. For weight matrix $w^1$ and $w^L$ with size $(D, h_1)$ and $(h_{L-1}, K)$, where $D$ and $K$ are number of input and output nodes, respectively, and $h_i$ is the number of nodes in the *i-th* hidden layer, then, elements $\{w^1(i \mod D, i); i = 1, \cdots, h_1\}$ and $\{w^L(i, i \mod K); i = 1, \cdots, h_{L-1}\}$ are selected as red weights, respectively.

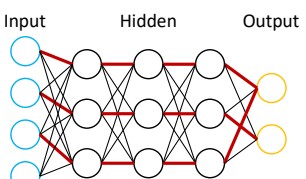

Figure 7: An example of feedforward ReLU network.

Besides, all methods can be easily applied to CNN. As we all know, a filter in CNN transforms a input feature map to the output feature map. Thus, each feature map can be regarded as a hidden node in MLP, and then the filter connects two feature maps in CNN corresponds to the weight connects two hidden nodes in MLP. Besides, the method to identify red weights is similar with that for MLP unless the diagonal elements of the 4-triple of the weight matrices are selected. Suppose the 4-triple of weight matrices is $[64, 64, 3, 3]$ where the first two elements denote the number of

input channels and output channels, the last two elements denote the size of filter. The element $[0, 0, 1, 1], [1, 1, 1, 1], [2, 2, 1, 1], \cdots, [63, 63, 1, 1]$ are selected to be red weights. In this way, the re-parameterization method in Theorem 3.1 is easy to extend to CNN by rescaling the weights using the selected red weights.

### D.2 Other Choices on Re-parameterization

In this section, we discuss other choices on re-parameterization and the influence. The re-parameterization method using path-values showed in the proof of Theorem 3.1 is not unique. For example, we can obtain another kind of re-parameterization, via multiplying and dividing the incoming and outgoing weights of $O_1^1$ by $v_{11}^1$ in step 4 of Figure.3. This will not influence analyses much in this paper for the following two reasons. First, whatever re-parameterization method used, each re-parameterized network can serve as a sufficient representation for ReLU networks in the path space, because the outputs of the network will keep unchanged for any input after re-parameterization. Second, our studies are based on Theorem 3.1, which can be generalized to other re-parameterization methods.

### D.3 Relationship between Re-parameterization and Optimization Process

In this section, we show how the network should be optimized after it has been re-parameterized. Specifically, after re-parameterization, we can optimize the loss function according to gradients with respect to the path-values using the optimization algorithm such as SGD as follows. First, we denote the parameter matrix of the re-parameterized network at the *l-th* layer as $W_l'$, whose elements are constant, path-value, and ration of path-values. Next, we can use back propagation to obtain the gradient of $W_l'$. According to the chain rule, for the path-values that will not appear in the denominator, its gradient can be calculated as:

$$\nabla_{v_{jk}^l} \mathcal{L} = \frac{\nabla_{w_{jk}^l} \mathcal{L}}{v_{kk}^1}, \tag{22}$$

and for the path-values that will appear in the denominator, its gradient can be calculated as:

$$\nabla_{v_{kk}^1} \mathcal{L} = \nabla_{w_{kk}^1} \mathcal{L} - \frac{\sum_{j=1, k=1}^d \nabla_{w_{jk}^l} \mathcal{L}}{v_{kk}^1} \tag{23}$$

Then, we can derive the update rule of SGD in path space mentioned in Section C. Here, please note that the update rules derived from re-parameterization is consistent with that provided in (Meng et al., 2019), and re-parameterization provides a much simpler and straightforward way to explain the complex update rules.

Based on the process above, we now provide some analyses on the magnitude of parameter matrices. Specifically, for the *2nd* term in Eq. 23, the numerator contains many terms, which tends to be larger than the gradient of other paths, and thus, the denominator, i.e., $v_{kk}^1, (k = 1, \cdots, d)$ is initialized to be large in practice to match the magnitude of the numerator. Therefore, for the parameter matrix $W_l'$, $v_{kk}^1, (k = 1, \cdots, d)$ appears in the denominator for most of its elements, and hence, these elements are initialized to be much smaller than 1, which supports analyses in Theorem 3.2.

## E   Full Algorithm for Optimizing ReLU Networks with $\mathcal{P}$-BN

In Algorithm 1, we show the full algorithm for optimizing ReLU networks with $\mathcal{P}$-BN in the path space. Besides, please check more details of $PathOptimizer$ in Section 2.2 and Appendix C.

## F   Details on Experiments

### F.1   Experimental Setting Details

In particular, following the settings in Meng et al. (2019), we apply $\mathcal{G}$-SGD or $\mathcal{P}$-BN with residual blocks, because there is no positive scaling invariance across residual blocks. In addition, we introduce a coefficient $\lambda(\lambda < 1)$ for $\bar{\bar{z}}_j^{l-1}(x^i)$ in Eq. 5 for experiments of ResNet, to prevent the output exploding during the forward process, and $\lambda$ for all layers in ResNet is set to be 0.1.

---

**Algorithm 1** Optimization Algorithm in the Path Space of ReLU networks with $\mathcal{P}$-BN.

---

**Require:** A mini-batch of inputs: $\mathcal{B} = \{x^1, \cdots, x^B\}$, initialization $w^{(0)}$, and $PathOptimizer$.
  **for** $t = 0, 1, \cdots, T$ **do**
    **Forward Process**
    1. Normalize the output in the *1st* layer in the same way as the conventional BN.
    **for** $l = 2, \cdots, L$ **do**
      2. Normalize the partial weighted summation by using the conventional BN according to Eq. 3.
      3. Calculate the mean and standard deviation of the partial weighted summation according to Eq. 4.
      4. Add the excluded term into the normalized partial weighted summation according to Eq. 5.
    **end for**
    5. Calculate the value of loss function $l(w^{(t)})$ by using the final output.
    **Backward Process**
    6. Calculate the gradient of $w^{(t)}$: $\nabla_w l(w^{(t)}) \leftarrow$ Back Propagation $(l(w^{(t)}); w^{(t)})$
    **Update Rule of Path Optimizer**
    7. $w^{(t+1)} = PathOptimizer(w^{(t)}, \nabla_w l(w^{(t)}))$.
  **end for**
**Ensure:** $w^{(T)}$.

---

Here we describe pre-processing steps, which is same as He et al. (2016). For CIFAR-10 and CIFAR-100, we randomly crop the input image by size of 32 with padding size of 4, and normalize every pixel value to $[0, 1]$. Then the random horizontal flipping to the image is applied. For ImageNet, we randomly crop the input image by size of 224, and normalize every pixel value to $[0, 1]$. Then the random horizontal flipping to the image is also applied.

We use 1 NVIDIA Tesla P100 or P40 GPU to run the experiments on CIFAR, and use 4 NVIDIA Tesla P100 or P40 GPUs in one machine to run the experiments on ImageNet. All experiments are averaged over 5 runs with different random seeds, and PyTorch (Paszke et al.) is used for implementation.

Here we list hyper-parameters used for CIFAR-10 and CIFAR-100. As for the hyper-parameters of methods mentioned in their corresponding papers, i.e., SGD+BN (He et al., 2016) and ($\mathcal{G}$-SGD)+BN(wnorm) (Meng et al., 2019), we use the same settings as their original ones. Specifically, for SGD+BN, the initial learning rate is set to 0.1, and for $\mathcal{G}$-SGD+BN (wnorm), the initial learning rate is set to 1.0. Besides, we tune hyper-parameters for ($\mathcal{G}$-SGD)+BN and the proposed ($\mathcal{G}$-SGD)+($\mathcal{P}$-BN). Specifically, the initial learning rate is searched from $\{0.1, 0.2, 0.5, 1.0\}$. Then, we use the method proposed in (Zheng et al., 2018) as the weight decay in the path space, and set the coefficient of the weight decay to $1 \times 10^{-4}$, for these two methods. Besides, we use the SGD without momentum in our experiments, because the way to utilize momentum in the path space remains unclear now. Moreover, for all models and algorithms, the mini-batch size is set to be 128 and the training process is conducted for 64k iterations. The learning rates are multiplied by 0.1 after 32k and 48k iterations in all experiments, and the coefficient of weight decay for methods is set to be 0.0001.

Here we list hyper-parameters used for ImageNet. For all experiments here, the mini-batch size is set to be 256 and the hyper-parameter for weight decay is set to be 0.0001 He et al. (2016). For SGD+BN, the initial learning rate is set to 0.1 He et al. (2016). For $\mathcal{G}$-SGD+$\mathcal{P}$-BN, we only tune the initial learning rate from $\{0.1, 0.2, 0.5\}$. The training process is conducted for 90 epochs, and the learning rate is multiplied by 0.1 after 30 and 60 epochs.

As for the stacked CNN used for observations in Section 5.2, we use the following settings: kernel size 3, stride 1, padding 1, and no bias. For CNN, the network width is the the number of channels, which is set to 128 for all hidden layers.

## F.2 SUPPLEMENTARY RESULTS

In Fig. 8, plot the training curve and test accuracy of 50-layer ResNet. Here, we add other baseline, i.e., SGD+$\mathcal{P}$-BN, in which we use SGD to train the network with $\mathcal{P}$-BN, and in Table 4, we show

| Dataset | Method | PlainNet | | ResNet | | |
|---|---|---|---|---|---|---|
| | | 18 | 34 | 18 | 34 | 50 |
| CIFAR-10 | SGD+$\mathcal{P}$-BN | 7.00% ($\pm$0.08) | 7.68% ($\pm$0.21) | 6.78% ($\pm$0.15) | 6.57% ($\pm$0.06) | 6.47% ($\pm$0.23) |
| CIFAR-100 | SGD+$\mathcal{P}$-BN | 28.57% ($\pm$0.46) | 32.98% ($\pm$0.81) | 26.71% ($\pm$0.22) | 26.80% ($\pm$0.51) | 25.68% ($\pm$0.28) |

Table 4: Test error rate on CIFAR-10 and CIFAR-100 for SGD+$\mathcal{P}$-BN.

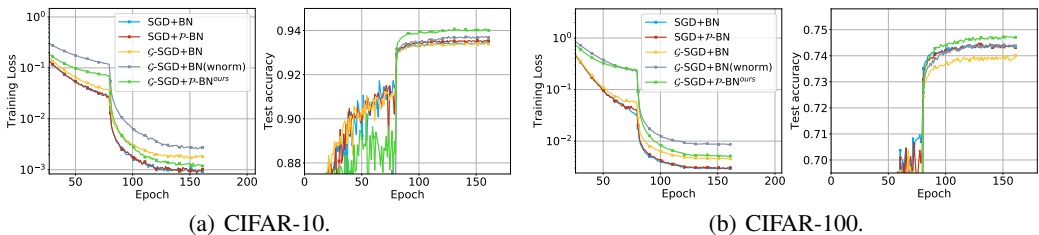

(a) CIFAR-10.
(b) CIFAR-100.

Figure 8: Training curves and test accuracy for training ResNet of 50 layers on CIFAR.

the test error for SGD+$\mathcal{P}$-BN. We can observe that $\mathcal{G}$-SGD+$\mathcal{P}$-BN outperforms SGD+$\mathcal{P}$-BN, which demonstrate the benefits of optimizing in the path space.

### F.3 EXPERIMENTS FOR PATH-SGD

As mentioned in Section 2.2, Path-SGD (Neyshabur et al., 2015a) is another PSI-related optimization algorithm, but not the optimizer in the path space.

*First*, we introduce the network structure and some basic settings. Here, because Path-SGD is computationally cost for CNN, we follow the experimental settings in (Neyshabur et al., 2015a), and use an MLP of 5 layers, whose number hidden nodes in each layer is 4000, and the conventional BN or $\mathcal{P}$-BN is added at the end of every hidden layer. Besides, we use CIFAR-10 as the training data, and the batch size is set to 64. Moreover, all experiments are run for 150 epochs, and results are obtained by averaging over 5 runs. The only hyper-parameter needs to be tuned is learning rate, which is searched from $\{1, 5\}^{-\{1,2,3\}}$.

| Method | Validation Error |
|---|---|
| SGD+BN | 35.30% ($\pm$0.54) |
| Path-SGD | 41.40% ($\pm$0.25) |
| Path-SGD+BN | 35.00% ($\pm$0.71) |
| Path-SGD+$\mathcal{P}$-BN | **34.45% ($\pm$0.29)** |

Table 5: Validation error for Path-SGD experiments.

*Then*, we show the setting details of 4 compared methods as follow:

- SGD+BN: We use SGD to train the network with the conventional BN.

- Path-SGD: We use Path-SGD to train the network without BN. [4]

- Path-SGD+BN: We use Path-SGD to train the network with the conventional BN.

- Path-SGD+$\mathcal{P}$-BN: We use Path-SGD to train the network with $\mathcal{P}$-BN.

In Table 5, we list the validation error of these 4 methods. Results show that: *(1)* Path-SGD+$\mathcal{P}$-BN outperforms other baselines, which well demonstrate the superiority of $\mathcal{P}$-BN. *(2)* Path-SGD+$\mathcal{P}$-BN and Path-SGD+BN perform better than SGD+BN, which show the benefits of optimizing in the path space.

---

[4] We do not apply this setting for ResNet and PlainNet, because the performance will be badly hurt, and training will fail when BN is removed.

### F.4 ANALYSIS OF COMPUTATIONAL COSTS

Different from the conventional BN, $\mathcal{P}$-BN is to identify the diagonal elements of weight matrices. For convolutional networks, the outputs of hidden nodes in MLP always correspond to the feature maps. Hence, on the one hand, if the sizes of feature maps in different layers are the same, and the stride length is $1$, we can then set the diagonal elements in weight matrices to $0$ and add skip connections for all layers (except the *1st* layer), which will not bring extra computational cost. On the other hand, if the stride length is larger than $1$, we can also set the diagonal elements in weight matrices to $0$, and additionally, use a fixed-weight convolutional layer, with only diagonal elements fixed to $1$ and others fixed to $0$. Thus, with the skip connection or the fixed-weight convolutional layer, the back propagation process can be easily implemented. Therefore, according to the above implementation, $\mathcal{P}$-BN will not bring much extra computational cost compared with the conventional BN. Statistically, in our experiments, the ratio of run time between $\mathcal{G}$-SGD+Path-BN and SGD+BN, is less than $4 : 3$, and the extra computational cost is mainly brought by the update of $\mathcal{G}$-SGD instead of $\mathcal{P}$-BN.

