# OpenReview forum: "P-BN: Towards Effective Batch Normalization in the Path Space"
_ICLR.cc/2020/Conference — Reject_

### Official Review · AnonReviewer2 · 2019-10-23
**Official Blind Review #2**

**Rating:** 3

**Review:**

This paper analyzes a reparametrization of the network that migrates from the weight space to the path space. It enables an easier way to understand the batch normalization (BN). Then the authors propose a variant of BN on the path space and empirically show better performance with the new proposal.

To study BN in the reparameterized space is well-intuited and a natural idea. Theorem 3.1 itself is interesting and has some value in understanding BN. However, the main contribution of the paper, i.e., the proposal of the P-BN, is not motivated enough. It is merely mentioned in the beginning of section 4 and it's not clear why this modification is better compared to conventional BN. This is not verified by theory either. By comparing theorem 3.2 and theorem 4.1, it seems P-BN even gives even worse upper bound of gradient norm.
Plus we don't actually care that much about the issue of gradient exploding since one could always do gradient clipping. The notorious gradient vanishing problem on the other hand, is not address in the theorems.
The formulation of the P-BN seems to be closely related to ResNet, since it sets aside the identity mapping and only normalizes on the other part. It would be better to have some discussions.
Also, the reparameterization and P-BN seems only to apply to fully connected layer from Eqn. (3-5) where they are proposed, but the experiments applies to ResNet. It would be better to describe the method in a broader sense. How would you do this P-BN in more complicated networks?

Finally, it's very unclear to me the value of Theorem 3.1 and the proof that takes almost one page in the main context. The assumption of diagonal elements of matrix w to be all positive is very restrictive and simply removes the effect of ReLU activations.

Therefore I think the paper has some room for improvement and is not very suitable for publication right now.







**Experience Assessment:**

I have published one or two papers in this area.

**Review Assessment: Checking Correctness Of Derivations And Theory:**

I assessed the sensibility of the derivations and theory.

**Review Assessment: Checking Correctness Of Experiments:**

I assessed the sensibility of the experiments.

**Review Assessment: Thoroughness In Paper Reading:**

I read the paper at least twice and used my best judgement in assessing the paper.

---

> ### Author Response · Authors · 2019-11-10
> **To Reviewer #2**
>
> Thanks for your comments. The following is our responses.
>
> Q1: “By comparing theorem 3.2 and theorem 4.1, it seems P-BN even gives even worse upper bound of gradient norm.”
> A1: This is not a worse upper bound. Theorem 3.2 and 4.1 provide the norm of gradient w.r.t outputs in every layer, when conventional BN and P-BN are applied to re-parameterized networks, respectively. Here, larger gradient norm means greater gradient exploding, so the larger one is the worse one. Please note that the diagonal elements of $\hat{W}_s$ in Theorem 3.2 are equal to 1 and in Theorem 4.1, we have stated that $\hat{W}’_s=\hat{W}_s-I$. Thus, theorem 4.1 demonstrates that the gradient exploding problem will be weaken by P-BN, i.e., the gradient norm can become smaller after applying P-BN, because the identical matrix is separated from $W’_s$, and the variance term $\|\sigma^{s,/}\|$ becomes larger. We have provided a comparison with clearer description in Corollary 4.2 in our new version.
>
> Q2: “we don't actually care that much about the issue of gradient exploding since one could always do gradient clipping.”
> A2: Gradient clipping is a trick which needs tuning hyper-parameters, and here we want to get rid of this trick and find a well-performed design for the problem. This paper aims to propose a suitable BN method in path space to ensure stable gradient propagation, which is more fundamental and have theoretical guarantee. It is unfair to criticize this paper’s significance to state that gradient exploding is not an important issue.
>
> Q3: “The formulation of the P-BN seems to be closely related to ResNet …”
> A3: We have some following differences. Frist, our motivation is quite different, as we start from the path space, while ResNet is not motivated by the new parameter space. Second, the novelty of ResNet is adding a skip-connection, but the identical connection is naturally exists in the path space, when the network is re-parameterized. Then, P-BN exclude the term related to the constant coefficient. Third, P-BN can also be used for ResNet since path space for ResNet is also established in previous work, which means that they are compatible.
>
> Q4: “It would be better to describe the method in a broader sense.”
> A4: We provided some details on CNN in Appendix D.1 in our original version. In that way, the number of hidden nodes in MLP corresponds to the number of channels in CNN, and CNN can be operated similarly with MLP. We have also added some additional details on CNN in the updated version.
>
> Q5: “The assumption of diagonal elements of matrix w to be all positive is very restrictive and simply removes the effect of ReLU activations.”
> A5: First, we lost the activation g in Theorem 3.1 and we have fixed it in our updated new version. We are sorry for this typo, and it may cause the misunderstanding that our theorem removes the effect of ReLU activations. In fact, ReLU activations still works after re-parameterization. Second, in the remark of theorem 3.1, we show that the positive constraint is not essential for proving theorem 3.1. Third, this constraint will not bring much influence on the model expressiveness, because the number of the constrained weights is tiny compared with the total weights, and according to experiment results in Meng et al., 2019 the practical performances are not harmed by this constrain. Actually, this constraint comes from the optimization algorithm in the path space (Meng et al., 2019), which is not introduced by our paper.
>
> We sincerely hope that we have addressed your concerns and you can reconsider your ratings after reading the responses and our updated version.

---

### Official Review · AnonReviewer3 · 2019-11-02
**Official Blind Review #3**

**Rating:** 3

**Review:**

Originality: The paper proposed a new Path-BatchNormalization in path space and compared the proposed method with traditional CNN with BN.

Quality: The theoretical part is messy but intuitive. Also, why P-BN helps path optimization is not clear in the paper. The experimental part is not convincing. All CNN with BN networks have much lower accuracy than people reported, e.g. https://pytorch.org/docs/stable/torchvision/models.html for ResNet on ImageNet.

Clarity: The written is not clear enough. It is not easy to imagine how the re-parameterization works on CNNs since the kernel is applied over the entire image ("hidden activations").

Significance:
See above.

**Experience Assessment:**

I do not know much about this area.

**Review Assessment: Checking Correctness Of Derivations And Theory:**

I assessed the sensibility of the derivations and theory.

**Review Assessment: Checking Correctness Of Experiments:**

I carefully checked the experiments.

**Review Assessment: Thoroughness In Paper Reading:**

I read the paper at least twice and used my best judgement in assessing the paper.

---

> ### Author Response · Authors · 2019-11-10
> **To Reviewer #3**
>
> Thanks for your suggestions. The following is our responses.
>
> Q1: “why P-BN helps path optimization is not clear in the paper.”
> A1: The reason that P-BN helps path optimization is described in section 3.2 and section 4. Specifically, in theorem 3.2, we demonstrate that gradients explode along network depth (layer index), because the variance term $\|\sigma^s\|$ is less than 1 and there is an identity matrix contained in $\hat{W}_s$ (please note that the diagonal elements of $\hat{W}_s$ in Theorem 3.2 are equal to 1 and in Theorem 4.1, we have stated that $\hat{W}’_s=\hat{W}_s-I$). Therefore, we propose P-BN, which only normalize the terms related to the trained coefficients, and exclude the term related to the constant coefficient. Then, as shown in theorem 4.1, the gradient exploding problem will be weaken by P-BN, because the identical matrix (constant coefficients) is separated from W’_s, and the variance term $\|\sigma^{s,/}\|$ becomes larger. We have provided a comparison with clearer description in Corollary 4.2 in our new version.
>
> Q2: “The experimental part is not convincing.”
> A2: We use the SGD without momentum in our experiments, because the way to utilize momentum in the path space remains unclear now. Our experiments follow the experimental settings of the work (Meng et al., 2019). We have clarified it in Appendix F.1 (Experimental Setting Details) in our updated version.
>
> Q3: “It is not easy to imagine how the re-parameterization works on CNNs since the kernel is applied over the entire image ("hidden activations").”
> A3: We provided some details on CNN in Appendix D.1 in our original version. In that way, the number of hidden nodes in MLP corresponds to the number of channels in CNN, and CNN can be operated similarly with MLP. We have also added some descriptions about this in Appendix D.1 in our new version.
>
> We sincerely hope that we have addressed your concerns and you can reconsider your ratings.

---

### Official Review · AnonReviewer4 · 2019-11-05
**Official Blind Review #4**

**Rating:** 3

**Review:**

The proposal is an adapted batch normalization method for path regularization methods used in the optimization of neural networks. For neural networks with Relu activations, there exits a particular singularity structure, called positively
scale-invariant, which may slow optimization. In that regard, it is natural to remove these singularities by optimizing along invariant input-output paths. Yet, the paper does not motivate this type of regularization for batchnormalized nets. In fact, batch normalization naturally remedies this type of singularity since lengths of weights are trained separately from the direction of weights. Then, the authors motivate their novel batch-normalization to gradient exploding (/vanishing) which is a completely different issue.
I am not sure whether I understood the established theoretical results in this paper. Let start with Theorem 3.1: I am not sure about the statement of the theorem. Is this result for a linear net? I think for a Relu net, outputs need an additional scaling parameter that depends on all past hidden states (outputs). Theorem 3.2 and 4.1 do not seem informative to me. Authors are saying that if some terms in the established bound in Theorem 4.1 is small, then exploding gradient does not occur for their novel method. The same argument can be applied to the plain batchnorm result in Theorem 3.2. For me, it is not clear to see the reason why the proposed method remedies the gradient exploding (/vanishing).


**Experience Assessment:**

I have published one or two papers in this area.

**Review Assessment: Checking Correctness Of Derivations And Theory:**

I assessed the sensibility of the derivations and theory.

**Review Assessment: Checking Correctness Of Experiments:**

I did not assess the experiments.

**Review Assessment: Thoroughness In Paper Reading:**

I made a quick assessment of this paper.

---

> ### Author Response · Authors · 2019-11-10
> **To Reviewer #4**
>
> Thank you for your comments. The following is our responses.
>
> Q1: “Let start with Theorem 3.1: I am not sure about the statement of the theorem. Is this result for a linear net? I think for a Relu net, outputs need an additional scaling parameter that depends on all past hidden states (outputs).”
> A1: No, all results in our paper are for non-linear neural networks with ReLU activations. We are sorry that we lost the activation function g in theorem 3.1, and we have fixed it in our new version.
>
> Q2: “Theorem 3.2 and 4.1 do not seem informative to me.”
> A2: Theorem 3.2 and 4.1 provide the norm of gradient w.r.t outputs in every layer, when conventional BN and P-BN are applied to re-parameterized networks, respectively. Please note that the diagonal elements of $\hat{W}_s$ in Theorem 3.2 are equal to 1 and in Theorem 4.1, we have stated that $\hat{W}’_s=\hat{W}_s-I$. Theorem 3.2 demonstrates that gradients explode along network depth (layer index), because the variance term $\|\sigma^s\|$ is less than 1 and there is an identity matrix contained in $\hat{W}_s$. Theorem 4.1 demonstrates that the gradient exploding problem will be weaken by P-BN, because the identical matrix is separated from $W’_s$, and the variance term $\|\sigma^{s,/}\|$ becomes larger. We provide a comparison with clearer description in Corollary 4.2 in our new version.
>
> Q3: “In fact, batch normalization naturally remedies this type of singularity since lengths of weights are trained separately from the direction of weights.”
> A3: Batch Normalization cannot fully remedy this type of singularity. Batch normalization can keep the outputs unchanged when weights in one layer and its successive layer is multiplied and divided by a positive constant. However, the gradients w.r.t weights are changed by such rescaling operation because the Lipschitz constants w.r.t weights at different layers have changed. An intuitive explanation is that if a BN network whose weights at different layers have unbalanced magnitudes, stochastic gradient descent will suffer from such “unbalanced” scale of weights. On the other hand, gradients can still keep unchanged when the network is optimized in the path space. Thus, studying batch normalization in the path space is important.
>
> We hope that we have answered your questions and addressed your concerns. We also hope that you can reconsider your ratings.

---

### Author Response · Authors · 2019-11-10
**Summary of the new version**

Thanks for all reviewers and their comments, which are helpful for us. We have uploaded a new version of our paper, and the main changes include:
1. We have fixed a typo in Theorem 3.1 by adding the activation g.
2. We have added Corollary 4.2.
3. We have added some more descriptions about CNN in Appendix D.1.

---

### Decision · Program_Chairs · 2019-12-19

**Decision:**

Reject

**Comment:**

This paper addresses the extension of path-space-based SGD (which has some previously-acknowledged advantages over traditional weight-space SGD) to handle batch normalization. Given the success of BN in traditional settings, this is a reasonable scenario to consider.  The analysis and algorithm development involved exploits a reparameterization process to transition from the weight space to the path space.  Empirical tests are then conducted on CIFAR and ImageNet.

Overall, there was a consensus among reviewers to reject this paper, and the AC did not find sufficient justification to overrule this consensus.  Note that some of the negative feedback was likely due, at least in part, to unclear aspects of the paper, an issue either explicitly stated or implied by all reviewers.  While obviously some revisions were made, at this point it seems that a new round of review is required to reevaluate the contribution and ensure that it is properly appreciated.